# EGGS-PTP: An Expander-Graph Guided Structured Post-training Pruning Method for Large Language Models

## Abstract

As Large Language Models (LLMs) become more widely adopted and scale up in size, the computational and memory challenges involved in deploying these massive foundation models have grown increasingly severe. This underscores the urgent need to develop more efficient model variants. Faced with this challenge, the present work introduces EGGS-PTP: an Expander-Graph Guided Structured Post-training Pruning method. The proposed approach leverages graph theory to guide the design of N:M structured pruning, effectively reducing model size and computational demands. By incorporating concepts from expander graphs, EGGS-PTP ensures information flow within the pruned network, preserving essential model functionality. Extensive numerical experiments demonstrate that EGGS-PTP not only achieves significant acceleration and memory savings due to structured sparsity but also outperforms existing structured pruning techniques in terms of accuracy across various LLMs.

## 1 Introduction

The rapid advancement in Large Language Models (LLMs) has revolutionized natural language processing (NLP), achieving remarkable performance across a wide range of language tasks. However, this progress comes with significant computational and memory demands, which poses substantial challenges for deploying these models in resource-constrained environments. As LLMs continue to scale, the development of efficient compression techniques becomes increasingly critical to democratize their usage and improve accessibility. To address these challenges, various approaches have been proposed, including model quantization (Bai et al., 2020; Frantar et al., 2022; Lin et al., 2024; Xiao et al., 2023), and model pruning (Hassibi et al., 1993; LeCun et al., 1989; Mocanu et al., 2018; Sun et al., 2023; Zhang et al., 2024).

In particular, model pruning has emerged as a prominent approach to reduce the size of neural networks by removing less critical weights while maintaining performance (Zhu et al., 2024). Among these methods, post-training pruning (PTP) stands out for its simplicity and efficiency (Bai et al., 2024; Frantar & Alistarh, 2023; Lu et al., 2024; Sun et al., 2023; Wang et al., 2024; Zafrir et al., 2021; Zhang et al., 2024), as it operates directly on pre-trained models without the need for extensive retraining. Despite its potential, existing pruning methods such as Wanda pruning (Sun et al., 2023) and RIA pruning (Zhang et al., 2024) struggle to maintain performance under structured sparsity constraints such as the $N : M$ sparsity pattern—a structure specifically designed for hardware acceleration on GPUs (Mishra et al., 2021).

In this paper, we introduce EGGS-PTP, an Expander-Graph Guided Structured Post-training Pruning method for LLMs. The proposed technique combines the $N : M$ sparsity constraint with the topological advantages of expander graphs, whose sparse yet highly connected structure facilitates robust information flow even under substantial pruning (Hoang & Liu, 2023; Prabhu et al., 2018; Spielman, 2007). To effectively preserve the most critical weights, EGGS-PTP further incorporates the principle of Relative Importance and Activations (**RIA**) (Zhang et al., 2024). Built upon the integration of these techniques, EGGS-PTP achieves a balance between computational efficiency and model performance.

Overall, our contributions can be summarized as follows:

- We propose EGGS-PTP, a novel post-training structured pruning framework inspired by expander graph theory to enable efficient and effective compression of LLMs.

- EGGS-PTP consists of two pruning strategies: importance-aware and structure-aware pruning, which focus on retaining weights with higher *quantitative* and *qualitative* importance, respectively. The two strategies with specific emphasis enable the proposed algorithm to preserve important weights and maintain information flow at the same time.

- The proposed EGGS-PTP also ensures $N : M$ sparsity that aligns with modern hardware acceleration constraints.

- Extensive experiments demonstrate that EGGS-PTP consistently outperforms existing structured pruning methods in both accuracy and efficiency.

## 2  RELATED WORKS

### 2.1  NEURAL NETWORK PRUNING

To reduce computational costs and memory usage without significantly compromising performance, pruning has become a widely adopted technique for compressing deep neural networks. Pre-training pruning removes weights before training, which is lightweight but often results in suboptimal performance and requires retraining (Hoang & Liu, 2023; Lee et al., 2018; Wang et al., 2020). During-training pruning introduces substantial training complexity and overhead (Huang et al., 2025; Xing et al., 2025). In contrast, post-training pruning is particularly appealing for large-scale LLMs, as it can yield high-performing sparse models with lower computational costs (Bai et al., 2024; Frantar & Alistarh, 2023; Lu et al., 2024; Sun et al., 2023; Wang et al., 2024; Zafrir et al., 2021; Zhang et al., 2024). For these reasons, this work adopts a post-training pruning framework for LLMs.

### 2.2  POST-TRAINING PRUNING

This work specifically focuses on post-training pruning (PTP), a promising approach for compressing LLMs due to its efficiency and simplicity (Bai et al., 2024; Frantar & Alistarh, 2023; Lu et al., 2024; Sun et al., 2023; Wang et al., 2024; Zafrir et al., 2021; Zhang et al., 2024). Unlike training-aware methods—such as sparse training (Mocanu et al., 2018; Sanh et al., 2020) or pruning-aware training (Han et al., 2015; Liu et al., 2021)—PTP operates directly on pre-trained models without requiring additional fine-tuning. This makes it a computationally efficient solution, especially suitable for resource-intensive LLMs. Classical approaches like Optimal Brain Damage (OBD) (LeCun et al., 1989) and Optimal Brain Surgeon (OBS) (Hassibi et al., 1993) leverage the Hessian matrix to identify and remove less important weights in Neural Networks. Building on these foundations, recent methods such as Wanda (Sun et al., 2023) and RIA (Zhang et al., 2024) have adapted pruning strategies to better suit LLMs, offering reduced complexity while maintaining competitive performance. Wanda improves one-shot pruning by incorporating input activations, while RIA introduces a novel metric to more effectively estimate weight importance. Despite these advancements, there remains substantial potential to further enhance both pruning accuracy and efficiency. To this end, we propose a novel PTP method guided by expander graphs, which improves information flow across layers and yields superior performance compared to existing approaches.

## 3  PRELIMINARIES AND PROBLEM FORMULATION

Post-training pruning (PTP) typically modifies a pre-trained model by directly altering its weight matrices, without requiring end-to-end fine-tuning. Consider a linear layer in LLM with an input $\mathbf{Z}^{\ell-1} \in \mathbb{R}^{F_{\ell-1}}$, an output $\mathbf{Z}^{\ell} \in \mathbb{R}^{F_{\ell}}$, and a weight matrix $\mathbf{W}^{\ell} \in \mathbb{R}^{F_{\ell} \times F_{\ell-1}}$, where $F_{\ell}$ and $F_{\ell-1}$ denote the number of output and input channels, respectively. Hence, the operation in the $\ell$-th layer can be represented as

$$\mathbf{Z}^{\ell} = \mathbf{W}^{\ell}\mathbf{Z}^{\ell-1}. \tag{1}$$

Given a pretrained LLM with weight matrices $\{\mathbf{W}^{\ell}\}$, we will adopt the $N : M$ sparsity constraint to enable efficient structured pruning (NVIDIA, 2020). To implement this, a binary pruning mask

$\mathbf{M}^\ell \in \{0,1\}^{F_\ell \times F_{\ell-1}}$ is constructed, where $\mathbf{M}^\ell_{ij} = 1$ indicates that the corresponding weight $\mathbf{W}^\ell_{ij}$ is retained, and $\mathbf{M}^\ell_{ij} = 0$ indicates that it is pruned. To satisfy the $N\!:\!M$ sparsity pattern, each group of $M$ elements in a row of $\mathbf{M}^\ell$ must contain exactly $N$ zeros and $M-N$ ones. The pruned weight matrix is then given by:

$$\tilde{\mathbf{W}}^\ell = \mathbf{W}^\ell \odot \mathbf{M}^\ell, \tag{2}$$

where $\odot$ denotes element-wise multiplication. Note that we omit the layer index $\ell$ in the following.

In this work, the goal is to take advantage of both $N\!:\!M$ sparsity and the strong connectivity of expander graphs to construct an effective binary mask $\mathbf{M}$, enabling efficient pruning while preserving robust model performance.

## 4 IMPORTANCE METRICS AND PREPROCESSING

We first introduce the importance metrics that will be used in developing the EGGS-PTP framework as well as the necessary preprocessing steps for the weight matrix.

### 4.1 IMPORTANCE METRICS

To guide the pruning, we rely on two importance metrics: Row Relative Importance (**RRI**) and Relative Importance with Activation (**RIA**). **RRI** provides a simple way to measure the contribution of each weight relative to other weights in the same output channel (row). **RIA** extends this idea by normalizing each weight with respect to both its output channel (row) and input channel (column), and then scaling the score by the corresponding input activation (Zhang et al., 2024). The detailed definitions of these two metrics are in Appendix A.

### 4.2 PRUNING GROUP PARTITION AND CHANNEL PERMUTATION

Note that $N\!:\!M$ pruning occurs row-wise within each **pruning group** of the weight matrix $\mathbf{W}$, where a pruning group under the $N\!:\!M$ sparsity constraint refers to the submatrix consisting the $kM + 1$ to $(k+1)M$ columns of $\mathbf{W}$, with $k \in \left[0, \frac{F_{\ell-1}}{M} - 1\right]$. However, such a pruning structure can lead to suboptimal pruning results when important weights are in the same pruning group, causing critical weights to be pruned mistakenly and trivial weights retained. To alleviate this, a channel permutation step is introduced to more evenly distribute important weights (Zhang et al., 2024).

Given **RIA** scores of all weights, we aggregate them column-wise to obtain a channel **RIA** for each input channel $\mathbf{W}_{*,j}$. The columns (input channels) are then sorted based on the channel **RIA** in descending order and reassigned to pruning groups using a round-robin allocation strategy proposed in (Zhang et al., 2024). Specifically, the $\frac{F_{\ell-1}}{M}$ input channels with the highest channel **RIA** are alternately allocated to the $\frac{F_{\ell-1}}{M}$ pruning groups. This process is iterated for $M$ times until all input channels are assigned. Channel permutation effectively reduces the likelihood that important weights are clustered in the same group. An example of channel permutation is given in Appendix B.

Note that all subsequent pruning steps are performed on the weight matrix after channel permutation. For notational simplicity, and with slight abuse of the notation, we will still use $\mathbf{W}$ as the permuted weight matrix in the following sections.

## 5 EXPANDER-GRAPH GUIDED STRUCTURED PRUNING

In this section, we present EGGS-PTP: an **E**xpander-**G**raph **G**uided **S**tructured **P**ost-**T**raining **P**runing Method for LLMs. It enjoys the computational efficiency resulting from structured pruning, while at the same time, maintains efficient information flow enabled by expander graphs.

The EGGS-PTP framework is designed to retain the important weights while also maintaining information flow in the pruned neural networks. Specifically, the information flow is preserved by fostering the expansion property in the underlying bipartite graphs of the pruned linear layers. The expansion property is particularly desirable in sparse neural networks because it guarantees that any small subset of neurons remains well-connected within the network even after pruning (Hsieh et al.,

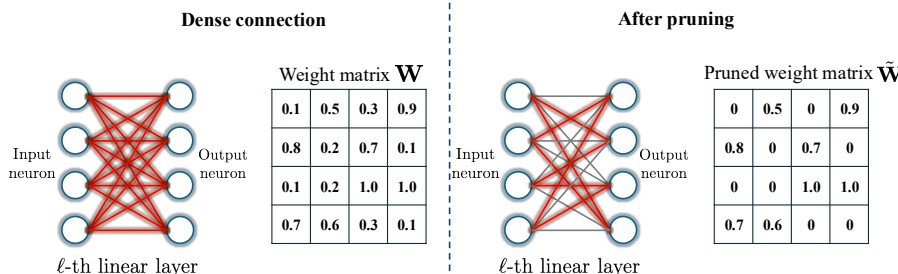

Figure 1: Example of modeling the $\ell$-th linear layer with $4$ input neurons and $4$ output neurons as a bipartite graph. The $4 \times 4$ weight matrix can be viewed as a submatrix of the $8 \times 8$ adjacency matrix of the bipartite graph, and pruning can be regarded as edge selection in the bipartite graph.

2024). In this way, the risk of performance degradation caused by channel corruption is mitigated. In the following subsections, we will further explain how to model the linear layer as a graph, as well as the design of our novel structured-pruning strategies.

## 5.1 GRAPH-BASED MODELING

Before diving into the pruning algorithm, we first introduce the definitions of bipartite graphs and two-sided expanders.

**Definition 1.** *A bipartite graph $\mathcal{G} = (\mathcal{I} \cup \mathcal{O}, \mathcal{E})$ is a graph in which the vertex set is divided into two disjoint subsets $\mathcal{I}$ and $\mathcal{O}$, such that every edge connects a vertex $v_i \in \mathcal{I}$ to a vertex $v_o \in \mathcal{O}$ (Asratian et al., 1998).*

**Definition 2.** *A bipartite graph $\mathcal{G} = (\mathcal{I} \cup \mathcal{O}, \mathcal{E})$ is a two-sided $(c, a_I, a_O)$-expander if there exists $c \in (0, 1)$ such that for $\forall \mathcal{T} \subseteq \mathcal{I}$ with $|\mathcal{T}| \leq c|\mathcal{I}|$, the neighborhood of $\mathcal{T}$, denoted as $\Gamma(\mathcal{T}) \subseteq \mathcal{O}$, satisfies (Hsieh et al., 2024):*

$$|\Gamma(\mathcal{T})| \geq a_I|\mathcal{T}|, \ \ a_I > 1. \tag{3}$$

*Symmetrically, for $\forall \mathcal{S} \subseteq \mathcal{O}$ with $|\mathcal{S}| \leq c|\mathcal{O}|$, the neighborhood $\Gamma(\mathcal{S}) \subseteq \mathcal{I}$ satisfies:*

$$|\Gamma(\mathcal{S})| \geq a_O|\mathcal{S}|, \ \ a_O > 1. \tag{4}$$

We model each linear layer as a bipartite graph $\mathcal{G} = (\mathcal{I} \cup \mathcal{O}, \mathcal{E})$, where $\mathcal{I}$ denotes the set of $|\mathcal{I}| = F_{\ell-1}$ input neurons and $\mathcal{O}$ denotes the set of $|\mathcal{O}| = F_\ell$ output neurons. Hence, the weight matrix $\mathbf{W} \in \mathbb{R}^{F_\ell \times F_{\ell-1}}$ can be viewed as the submatrix of the adjacency matrix of the bipartite graph $\mathcal{G}$, where $\mathbf{W}_{ij}$ denotes the edge weight between the vertex $v_i$ and $v_j$ (as illustrated in Figure 1). In this way, pruning can be considered as edge selection in the bipartite graph. Specifically, $\mathbf{M}_{ij} = 0$ means the corresponding edge is pruned, and hence $\tilde{\mathbf{W}}_{ij} = 0$. On the other hand, the weight is retained as $\tilde{\mathbf{W}}_{ij} = \mathbf{W}_{ij}$ if $\mathbf{M}_{ij} = 1$ (see Equation 2). In the case where the weights are pruned to ensure $N : M$ sparsity, each output neuron is henceforth guaranteed to have $M - N$ neighbors in each pruning group of $M$ input neurons.

In order to achieve $N : M$ sparsity while avoiding significant performance degradation, two requirements should be met:
**r1)** The retained edges should be selected such that the important edges are preserved.
**r2)** The resulting graph after pruning should remain "well-connected" to preserve information flow from layer to layer.

To meet requirement **r1)**, a simple solution in existing works (Zhang et al., 2024; Sun et al., 2023; Frantar & Alistarh, 2023) is first identifying the $M - N$ most important weights in each pruning group based on importance metrics, and directly setting the corresponding mask to $1$, and $0$ otherwise. This can be summarized as the Importance-aware Pruning detailed in the next subsection.

## 5.2 IMPORTANCE-AWARE PRUNING

This strategy aims to retain the top $M - N$ most important weights based on the **RIA** scores. Specifically, for a row in each pruning group, we retain the top $M - N$ weights with the highest **RIA**

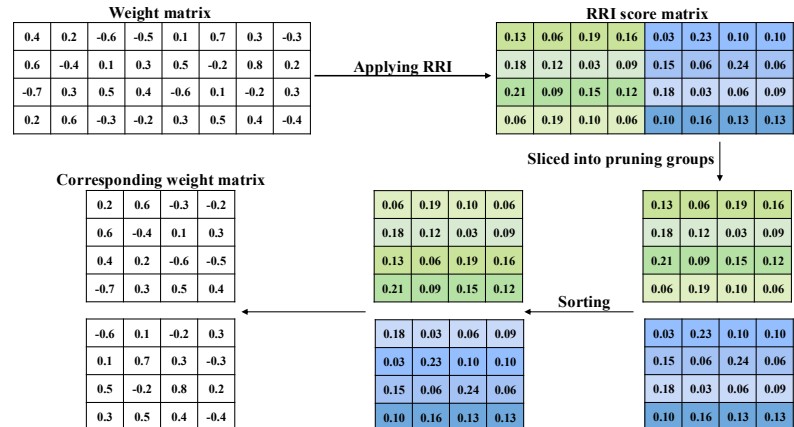

Figure 2: Example of applying importance-aware partitioning with $2:4$ sparsity on a weight matrix.

scores by setting the corresponding elements in $\mathbf{M}$ to 1, ensuring that the most influential connections are preserved. By focusing on the preservation of high-impact weights, this strategy helps maintain the model's performance after pruning.

Nevertheless, such pruning may not necessarily meet **r2)**. While each output neuron is guaranteed to have $M - N$ neighbors in each group of $M$ input neurons due to the definition of $N : M$ sparsity, the connectivity of input neurons can be arbitrary. In the case where all weights corresponding to an input channel/neuron have low importance scores, it is possible that most or even all of its edges are pruned. This causes input channel corruption and hinders information flow over the graph (Zhang et al., 2024). Note that channel corruption is harmful for LLMs since each channel typically encodes unique information critical to model performance. To tackle this challenge, a connectivity-aware pruning strategy will be introduced next to preserve the underlying connectivity.

### 5.3 CONNECTIVITY-AWARE PRUNING

The goal of connectivity-aware pruning is to preserve the connectivity, such that the resulting pruned sparse linear layers maintain information flow without channel corruption. The design of this strategy is inspired by the construction of expander graphs, and the expansion properties of the resulting graphs will be analyzed rigorously in the next section.

For each pruning group consisting of $M$ columns, the connectivity-aware pruning strategy is applied to multiple $M \times M$ blocks in two steps:

**Step 1: Diagonal selection.** To mitigate the channel corruption, we introduce the diagonal selection to ensure that each input and output neuron retains at least one connection after pruning. Instead of choosing a fixed diagonal, the proposed selection scheme seeks to retain large-magnitude weights, which are important for preserving the model performance. To this end, each block is first divided into $2 \times 2$ sub-blocks (i.e., quadrants). In each sub-block, the sum of weight magnitudes is computed along both the main diagonal and the anti-diagonal, and the diagonal with the larger sum is selected. The selected diagonals from the top-left and bottom-right sub-blocks are grouped as one pair, and those from the top-right and bottom-left as the other. We calculate the total sums of the selected diagonals for both pairs, and retain the pair with the larger sum by setting the corresponding entries in the binary pruning mask $\mathbf{M}$ to 1. Through this selection, each input/output channel is guaranteed to retain at least one connection after pruning, which helps preserve connectivity across input and output neurons and reduces the risk of channel corruption. An example of the diagonal selection is provided in Appendix C.

**Step 2: RIA-based top $M - N - 1$ selection.** Following the diagonal selection, which preserves one weight per output channel (row), to satisfy the $N : M$ sparsity constraint, an additional $M - N - 1$ weights are selected from each row based on their **RIA** scores. This step prioritizes weights that are critical to performance, thereby reducing the risk of performance degradation caused by pruning.

---

**Algorithm 1** EGGS-PTP Pruning

---

**Input**: Weight matrix $\mathbf{W} \in \mathbb{R}^{F_\ell \times F_{\ell-1}}$, sparsity constraint $N : M$, number of blocks $B$ for Connectivity-aware Pruning
**Output**: Mask $\mathbf{M} \in \{0, 1\}^{F_\ell \times F_{\ell-1}}$

1: Initialize $\mathbf{M} = \{0\}^{F_\ell \times F_{\ell-1}}$.
2: Compute **RIA** scores.
3: Channel permutation according to channel **RIA**.
4: Compute **RRI** scores.
5: Partition $\mathbf{W}$ into $\frac{F_{\ell-1}}{M}$ pruning groups.
6: **for** each pruning group **do**
7:     Compute row-wise aggregated **RRI** scores and sort the rows in ascending order.
8:     Divide the pruning group into $M \times M$ blocks.
9:     **for** each $M \times M$ block **do**
10:       **if** block index $< B$ **then**
11:         **Diagonal Selection**: select one diagonal (main or anti-diagonal) per quadrant based on magnitude sum, then retain the pair of diagonals with the higher sum, and set the corresponding $\mathbf{M}_{ij} = 1$.
12:         **RIA-based Selection**: retain additional $M - N - 1$ entries per row based on top **RIA** scores, and set the corresponding $\mathbf{M}_{ij} = 1$.
13:       **else**
14:         **RIA-based Selection**: retain top $M - N$ entries per row based on **RIA** scores, and set the corresponding $\mathbf{M}_{ij} = 1$
15:       **end if**
16:     **end for**
17: **end for**
18: **return M**

---

## 5.4 Importance-aware Partitioning

Note that the importance-aware pruning strategy focuses on retaining weights with higher *quantitative* importance, while the connectivity-aware strategy emphasizes the structure, and henceforth the *qualitative* importance of the weights. To achieve the best of the two worlds, we will use the importance-aware pruning strategy on the weights with higher quantitative importance to maximize the effect of importance-aware selection. Meanwhile, the connectivity-aware strategy will be applied to weights with lower quantitative importance, to solely focus on the structure of the graph. To this end, we develop a principled way to partition each pruning group into blocks of different importance.

Given the permuted weight matrix $\mathbf{W}$ obtained from channel permutation, the **RRI** scores. For each pruning group, we compute the row-wise sum of the corresponding **RRI** scores and sort the rows in ascending order according to the obtained sum. An example of the importance-aware partitioning process can be found in Figure 2. The connectivity-aware pruning will be applied to the $B$ blocks with the lowest aggregated **RRI** scores, where $B$ is a hyperparameter. Meanwhile, the importance-aware pruning will be applied on the rest rows with higher aggregated **RRI** scores. The simple yet effective partitioning mechanism helps identify blocks with different levels of importance within the pruning groups and therefore facilitates the choice of blocks for applying different pruning strategies mentioned above.

To summarize, the proposed EGGS-PTP framework achieves $N : M$ structured sparsity by two tailored pruning strategies. The importance-aware pruning strategy focuses on preserving weights with higher relative importance, while the connectivity-aware pruning aims to maintain graph connectivity. By combining them, the proposed framework can effectively preserve the important weights and maintain information flow within each linear layer. The overall pruning process is detailed in Algorithm 1.

## 6 Theoretical Analysis

In this section, we rigorously analyze the structural properties of the pruned linear layers obtained from the EGGS-PTP framework. In particular, we show that the bipartite graph formed within each pruned linear layer is a two-sided expander, see Definition 2. This is formalized in Lemma 1.

Table 1: Wikitext2 perplexity results under a $2\!:\!4$ sparsity pattern.

| Method | LLaMA2-7b | LLaMA2-13b | LLaMA3-8b | LLaMA-34b | LLaMA3.2-1b | OPT-1.3b |
|---|---|---|---|---|---|---|
| Dense | 5.47 | 4.88 | 6.13 | 5.58 | 9.75 | 14.23 |
| Magnitude | 37.76 | 8.88 | 2400 | 8.63 | 3289.17 | 377.86 |
| Wanda | 11.35 | 8.76 | 24.83 | 7.49 | 70.85 | 28.34 |
| RIA | 10.41 | 8.08 | 21.47 | 6.92 | 68.35 | 27.65 |
| EGGS-PTP (Ours) | **10.32** | **7.93** | **20.88** | **6.78** | **60.93** | **26.78** |

**Lemma 1.** *Let $\tilde{\mathcal{G}} := (\mathcal{I} \cup \mathcal{O}, \tilde{\mathcal{E}})$ be the bipartite graph corresponding to the pruned weight matrix obtained from the EGGS-PTP framework. Then $\tilde{\mathcal{G}}$ is a two-sided $(c, a_I, a_O)$-expander for $c \in (0, 1)$, with $a_I > 1$ and $a_O > 1$.*

*Proof.* Recall Definition 2, the pruned bipartite graph $\tilde{\mathcal{G}}$ is a two-sided $(c, a_I, a_O)$-expander if there exists $c \in (0, 1)$ such that $\forall \mathcal{T} \subseteq \mathcal{I}$ with $|\mathcal{T}| \leq c|\mathcal{I}|$, the neighborhood satisfies $|\Gamma(\mathcal{T})| \geq a_I|\mathcal{T}|$. Symmetrically, there exists $c \in (0, 1)$ such that $\forall \mathcal{S} \subseteq \mathcal{O}$ with $|\mathcal{S}| \leq c|\mathcal{O}|$, the neighborhood satisfies $|\Gamma(\mathcal{S})| \geq a_O|\mathcal{S}|$ .

To prove $a_I > 1$, we consider any subset $T \subseteq \mathcal{I}$. According to Algorithm 1, the repeated diagonal selection across $B$ blocks in each pruning group ensures that every input neuron $v_i \in \mathcal{I}$ retains at least $B$ connections, yielding:

$$|\Gamma(\mathcal{T})| \geq B. \tag{5}$$

Therefore, for any $\mathcal{T} \subseteq \mathcal{I}$ with $|\mathcal{T}| < B$, i.e., $|\mathcal{T}| \leq c_I|\mathcal{I}| = c_I F_{\ell-1}$ and $c_I \in (0, \frac{B}{F_{\ell-1}})$, we obtain

$$a_I > 1, \tag{6}$$

satisfying Equation 3. In order to prove $a_O > 1$, we consider a subset $\mathcal{S} \subseteq \mathcal{O}$. Under the $N\!:\!M$ sparsity, each output neuron is guaranteed to retain exactly $M - N$ connections within each pruning group. According to EGGS-PTP, the weight matrix is divided into $\frac{F_{\ell-1}}{M}$ pruning groups; thus, after pruning, each output node $v_o \in \mathcal{S} \subseteq \mathcal{O}$ has a degree of $deg(v_o) = \frac{F_{\ell-1}}{M}(M - N)$. Thus, we have

$$|\Gamma(\mathcal{S})| \geq \frac{F_{\ell-1}}{M}(M - N). \tag{7}$$

Therefore, for any $\mathcal{S} \subseteq \mathcal{O}$ with $|\mathcal{S}| < \frac{F_{\ell-1}}{M}(M - N)$, i.e., $|\mathcal{S}| \leq c_O|\mathcal{O}| = c_O F_\ell$ for some $c_O \in (0, \frac{F_{\ell-1}(M-N)}{F_\ell M})$ , we have

$$a_O > 1, \tag{8}$$

satisfying Equation 4. Combining Equation 6 and Equation 8, we conclude that $\tilde{\mathcal{G}}$ is a two-sided expander for $c \in (0, \min(\frac{B}{F_{\ell-1}}, \frac{F_{\ell-1}(M-N)}{F_\ell M}))$. □

Building on Lemma 1, we conclude that each pruned linear layer is a two-sided expander. Therefore, the linear layers pruned by the EGGS-PTP framework inherit the expansion property from both the input and output sides.

**Remark 1.** *Lemma 1 reveals that the hyperparameter $B$ influences the expansion property, with a larger value of $B$ leading to a better expansion property. However, increasing $B$ also reduces the number of blocks over which the importance-aware pruning strategy was applied. Hence, $B$ determines the tradeoff between importance-aware (quantitative) and connectivity-aware (qualitative) pruning. In practice, $B$ is a hyperparameter that can be selected based on cross-validation.*

## 7 EXPERIMENTS

In this section, we conducted comprehensive experiments to evaluate the effectiveness and efficiency of the proposed EGGS-PTP framework under various settings.

Table 2: Wikitext2 perplexity results under a $4:8$ sparsity pattern.

| Method | LLaMA2-7b | LLaMA2-13b | LLaMA3-8b | LLaMA-34b | LLaMA3.2-1b | OPT-1.3b |
|---|---|---|---|---|---|---|
| Dense | 5.47 | 4.88 | 6.13 | 5.58 | 9.75 | 14.25 |
| Magnitude | 15.91 | 7.32 | 181.48 | 7.77 | 843.26 | 220.94 |
| Wanda | 8.42 | 6.87 | 14.94 | 6.70 | 42.27 | 22.59 |
| RIA | 8.11 | 6.62 | 13.60 | 6.47 | 39.77 | 21.86 |
| EGGS-PTP (Ours) | **8.04** | **6.58** | **12.97** | **6.44** | **38.94** | **21.58** |

Table 3: Zero-shot performance results for LLaMA-34b using a $2:4$ sparsity pattern. $^*$ indicates performance exceeding dense baseline.

| Method | Hellaswag | BoolQ | ARC-C | MNLI | RTE | Average |
|---|---|---|---|---|---|---|
| Dense | 60.72 | 79.45 | 49.68 | 38.94 | 63.18 | 58.39 |
| Magni. | 54.57 | 69.79 | 44.14 | 29.15 | 55.67 | 51.26 |
| Wanda | 57.62 | 78.68 | 47.73 | **37.48** | 64.77$^*$ | 57.45 |
| RIA | 58.93 | 79.62$^*$ | 47.85 | 36.52 | 64.93$^*$ | 57.65 |
| EGGS-PTP (Ours) | **60.11** | **80.12**$^*$ | **48.40** | 37.32 | **65.21**$^*$ | **58.23** |

## 7.1 SETUP

We tested EGGS-PTP on a variety of models from the LLaMA2 (Touvron et al., 2023), LLaMA3 (Dubey et al., 2024), and OPT (Zhang et al., 2022) families. We used the HuggingFace transformers library and $4$ NVIDIA A4000 GPUs with 16GB of memory each. For each of the tested LLMs, we apply the same pruning method to all linear layers except for the embeddings and the head. We tested both the perplexity as well as the zero-shot performance of the models.

**Baselines:** We have compared our model to other PTP methods in the experiments. We use the magnitude pruning (Han et al., 2015), Wanda (Sun et al., 2023), and RIA as our naive baseline approaches (Zhang et al., 2024). For compatibility and fairness, we follow the calibration protocol used in prior work and adopt the same set of calibration data as SparseGPT, consisting of sampled sequences from the C4 training set (Raffel et al., 2020).

**Datasets:** Following previous works on LLM pruning Sun et al. (2023); Zhang et al. (2024), we evaluate the perplexity on Wikitext2 Merity et al. (2016). For zero-shot evaluation, we follow common practice and use Hellaswag, BoolQ, ARC-C, MNLI, and RTE, which cover commonsense reasoning, QA, science reasoning, and natural language inference.

To evaluate EGGS-PTP, we performed 1-fold cross-validation on the validation set by tuning the hyperparameter $B$ over the range $[2, 20]$. The value of $B$ that yielded the best validation performance was then applied to the test set.

## 7.2 PERPLEXITY RESULTS

As highlighted in Tables 1 and 2, EGGS-PTP consistently outperforms other PTP methods across both sparsity patterns and all evaluated models. Notably, under $2:4$ sparsity, EGGS-PTP achieves an average $6.66\%$ of improvement in performance compared to RIA, while RIA improves upon Wanda by $15.05\%$. Similarly, under the $4:8$ sparsity pattern, our method demonstrates a $3.86\%$ reduction in performance drop over RIA, whereas RIA outperforms Wanda by $12.54\%$. As observed, the difference in performance improvement between EGGS-PTP and RIA is smaller than that between RIA and Wanda. This is because as performance approaches the dense baseline, it becomes challenging to achieve a significant increase. The experiments also showcase that the optimal value of $B$ generally increases with model size. It is also essential to note that, despite the promising performance of our method, smaller dense models often rival larger pruned models. For example, the dense LLaMA2 7B obtains a lower perplexity than the EGGS-PTP pruned LLaMA2 13B. This can be attributed to the strictness of $N:M$ sparsity patterns, which, while enabling computational efficiency, impose

Table 4: Zero-shot performance results for LLaMA-34b using a $4:8$ sparsity. $*$ indicates performance exceeding dense baseline.

| Method | Hellaswag | BoolQ | ARC-C | MNLI | RTE | Average |
|---|---|---|---|---|---|---|
| Dense | 60.72 | 79.45 | 49.68 | 38.94 | 63.18 | 58.39 |
| Magni. | 56.20 | 73.54 | 45.23 | 31.35 | 57.03 | 52.67 |
| Wanda | 58.89 | 79.60$^*$ | 48.16 | **38.09** | 65.17$^*$ | 57.95 |
| RIA | 59.11 | 81.46$^*$ | 48.58 | 36.61 | 65.52$^*$ | 58.25 |
| EGGS-PTP (Ours) | **60.34** | **81.89**$^*$ | **49.21** | 37.94 | **66.01**$^*$ | **59.10** |

constraints that may limit the model's ability to fully retain its original performance. We further evaluate EGGS-PTP on LLaMA 65B and LLaMA2 70B, with results included in Appendix D. We also report Pile perplexity in Appendix E.

Table 5: Runtime of Wikitext2 perplexity on LLaMA2-13b model.

| | Dense | Magnitude | Wanda | Wanda 2:4 | RIA | RIA 2:4 | EGGS-PTP 2:4 |
|---|---|---|---|---|---|---|---|
| Runtime (s) | 52.14 | 51.51 | 51.21 | 31.79 | 51.48 | 31.91 | 31.82 |
| Speed-up | 1$\times$ | 1.01$\times$ | 1.02$\times$ | **1.64$\times$** | 1.01$\times$ | 1.63$\times$ | **1.64$\times$** |

### 7.3 ZERO-SHOT RESULTS

In Tables 3 and 4, we present the zero-shot performance of LLaMA 34B pruned by different PTP methods under $2:4$ and $4:8$ sparsity patterns. The results demonstrate that EGGS-PTP consistently outperforms existing methods across nearly all benchmarks. Notably, EGGS-PTP even surpasses the dense model on BoolQ and RTE, achieving $80.12$ on BoolQ compared to the dense model's $79.45$. This improvement can be attributed to the reduction of overfitting, a common issue in dense models. By masking less important weights, EGGS-PTP simplifies the model, effectively addressing overfitting and enhancing generalization. Taken together, the results on both Wikitext2 perplexity and zero-shot tasks demonstrate the effectiveness of the proposed EGGS-PTP framework. By integrating the importance-aware and the connectivity-aware pruning strategies, the framework preserves the model performance. Ablation studies in Appendix F further verify the contribution of each component of the proposed framework to the overall performance.

### 7.4 RUNTIME ANALYSIS

In Table 5, we present the runtime performance of computing perplexity on the Wikitext2 dataset using LLaMA2 13B. EGGS-PTP demonstrates a short runtime of $31.82$ seconds. Specifically, all PTP methods employing $2:4$ sparsity achieve nearly identical $1.64\times$ speedup over the dense model, demonstrating the efficiency of structured $N:M$ sparsity in accelerating inference. In contrast, unstructured pruning methods show minimal speedups. The corresponding pruning overhead is provided in Appendix G, which further highlights the practicality of EGGS-PTP.

## 8 CONCLUSION

This paper introduced EGGS-PTP, an Expander-Graph Guided Structured Post-Pruning Method for LLMs. By integrating the expander graph theory into the pruning process, EGGS-PTP ensures that the resulting pruned networks preserve strong connectivity and robust information flow. The method combines two complementary pruning strategies: importance-aware pruning, which retains weights with high RIA, and connectivity-aware pruning, which enforces expansion structure via diagonal selection. A principled partitioning mechanism is proposed to determine where each strategy is applied, striking a balance between the quantitative and qualitative importance. Experimental results demonstrate that EGGS-PTP outperforms existing structured pruning techniques in terms of both perplexity and zero-shot accuracy while ensuring efficiency. Overall, EGGS-PTP offers a principled and effective approach for compressing LLMs.

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

# A  IMPORTANCE METRICS

To effectively assess the weight importance for guiding the subsequent pruning process, we introduce two importance metrics: **RRI** and **RIA**. These metrics quantify how critical each weight is to model performance, enabling informed decisions during pruning.

**Row Relative Importance (RRI).**  **RRI** is used to quantify the importance of each weight with respect to the corresponding row. Formally, $\mathbf{RRI}_{ij}$ is obtained by normalizing the magnitude of each weight $\mathbf{W}_{ij}$ with the sum of the absolute values of the weights in the corresponding row:

$$\mathbf{RRI}_{ij} = \frac{|\mathbf{W}_{ij}|}{\sum_{k=1}^{F_{\ell-1}} |\mathbf{W}_{ik}|}. \tag{9}$$

The computational complexity of **RRI** is $\mathcal{O}(F_\ell \times F_{\ell-1})$.

**Relative Importance and Activation (RIA).**  **RIA** is a metric introduced to combine the relative importance of the neural network weights and activation information (Zhang et al., 2024). To quantify the significance of each weight $\mathbf{W}_{ij}$ relative to the corresponding row $i$ (output channel) and column $j$ (input channel), the weights are normalized using the absolute sum of the corresponding column and row. Furthermore, to incorporate the input activation, which provides additional information about the actual contribution of weights to the network's outputs, the $\ell_2$ norm of the corresponding input $\|\mathbf{Z}_j^{\ell-1}\|$ is utilized. Hence, the **RIA** score corresponding to the weight $\mathbf{W}_{ij}$ can be written as:

$$\mathbf{RIA}_{ij} = \left( \underbrace{\frac{|\mathbf{W}_{ij}|}{\sum_{k=1}^{F_{\ell-1}} |\mathbf{W}_{ik}|}}_{(a)} + \underbrace{\frac{|\mathbf{W}_{ij}|}{\sum_{k=1}^{F_\ell} |\mathbf{W}_{kj}|}}_{(b)} \right) \underbrace{\|\mathbf{Z}_j^{\ell-1}\|_2^\alpha}_{\text{input activation}}, \tag{10}$$

where (a) denotes the relative importance with respect to (w.r.t.) row $i$ (see Equation 9), and (b) represents the relative importance w.r.t. column $j$. The parameter $\alpha$ controls the strength of activations. The computational complexity of **RIA** is $\mathcal{O}(F_\ell \times F_{\ell-1})$.

# B  CHANNEL PERMUTATION

Figure 3 illustrates an example in which $2:4$ sparsity is applied to a $4 \times 8$ weight matrix. Under a $2:4$ sparsity pattern with 4 output channels and 8 input channels, the weight matrix is divided into 2 pruning groups. The top 1 and top 2 input channels are assigned to group 1 and 2, respectively, and the rest of the channels are distributed to the groups following the same strategy.

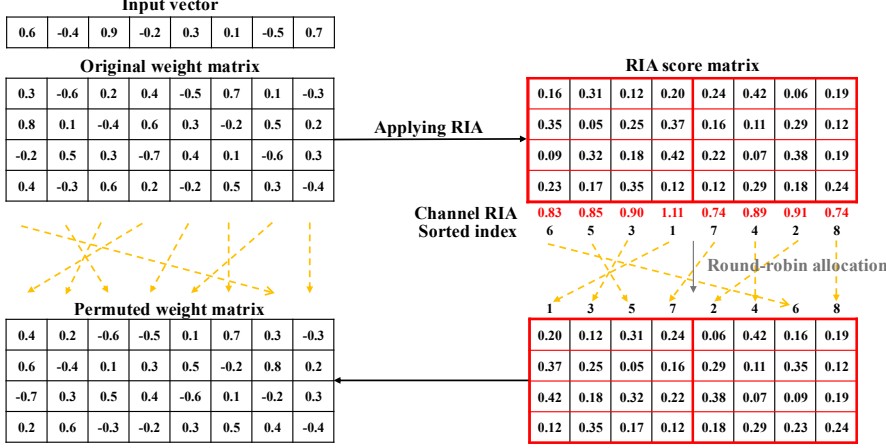

Figure 3: Example of channel permutation on a $4 \times 8$ weight matrix based on $2:4$ sparsity.

## C   DIAGONAL SELECTION

To facilitate the information flow across linear layers, we proposed a diagonal selection strategy that guarantees that each input and output neuron retains at least one connection. Each $M \times M$ block is first divided into multiple $2 \times 2$ or $4 \times 4$ sub-blocks (depending on $M$). Within each quadrant, we compute the sum of magnitudes along both the main diagonal and the anti-diagonal, and retain the diagonal with the larger sum. The diagonals selected from the top-left and bottom-right quadrants form one candidate pair, while those from the top-right and bottom-left quadrants form another. The pair with the higher total magnitude is then chosen, and the corresponding weights are retained. An example is provided in Figure 4, where blue values highlight the selected diagonals within each quadrant, and bold values indicate the retained pair.

| 0.65 | **0.83** | 0.37 | 0.31 |
|------|----------|------|------|
| **0.90** | 0.97 | 0.63 | 0.27 |
| 0.53 | 0.08 | 0.47 | **0.82** |
| 0.81 | 0.50 | **0.81** | 0.28 |

Figure 4: Example of the diagonal selection.

## D   SCALABILITY

To evaluate the scalability of EGGS-PTP, we extend our experiments to LLaMA 65B and LLaMA2 70B under the $2:4$ sparsity pattern. Table 6 reports perplexity on Wikitext2. EGGS-PTP consistently achieves the lowest perplexity among all baselines, further demonstrating its effectiveness.

Table 6: Wikitext2 perplexity results of LLaMA2-70b and LLaMA-65b under a $2:4$ sparsity pattern.

| Method | Dense | Magnitude | Wanda | RIA | EGGS-PTP (Ours) |
|--------|-------|-----------|-------|-----|------------------|
| LLaMA2-70b | 3.57 | 5.69 | 5.72 | 5.43 | **4.24** |
| LLaMA-65b | 3.82 | 6.02 | 6.03 | 5.83 | **4.96** |

## E   ADDITIONAL PERPLEXITY RESULTS

To complement the Wikitext2 results shown in Table 1 and Table 2, we additionally evaluate perplexity on the validation split of Pile Gao et al. (2020). All models are pruned under the $2:4$ sparsity pattern, and the results are summarized in Table 7. Among the pruning methods, EGGS-PTP achieves the lowest perplexity across every model scale.

Table 7: Pile perplexity results under a $2:4$ sparsity pattern.

| Method | LLaMA2-7b | LLaMA2-13b | LLaMA3-8b | LLaMA-34b | LLaMA3.2-1b | OPT-1.3b |
|--------|-----------|------------|-----------|-----------|-------------|----------|
| Dense | 4.92 | 4.60 | 5.68 | 4.84 | 7.85 | 7.58 |
| Magnitude | 19.20 | 9.19 | 149.14 | 6.94 | 912.45 | 356.61 |
| Wanda | 12.08 | 9.60 | 21.96 | 6.26 | 66.14 | 19.08 |
| RIA | 10.71 | 8.53 | 21.14 | 6.05 | 69.59 | 18.40 |
| EGGS-PTP (Ours) | **10.04** | **8.51** | **19.48** | **6.01** | **61.35** | **17.93** |

# F ABLATION STUDIES

To investigate the contribution of each component in the EGGS-PTP framework, we conducted an ablation study on LLaMA3.2 1B and OPT 1.3B under the $2:4$ sparsity pattern, with results summarized in Table 8. We observe that removing any individual module leads to performance degradation, showing that each design choice contributes to pruning effectiveness. Notably, the EGGS-PTP framework consistently achieves the best results, which demonstrates the complementary nature of the importance-aware pruning and the connectivity-aware pruning.

Table 8: Ablation study results (Wikitext2 perplexity) under $2:4$ sparsity pattern on LLaMA3.2-1b and OPT-1.3b.

| Method | LLaMA3.2-1b | OPT-1.3b |
|---|---|---|
| w/o Channel Permutation | 92.84 | 27.56 |
| w/o Importance-aware Pruning | 157.56 | 42.34 |
| w/o Connectivity-aware Pruning | 68.35 | 27.65 |
| EGGS-PTP (Ours) | **60.93** | **26.78** |

## G PRUNING OVERHEAD

Table 9 shows the pruning overhead for OPT 1.3B. Although the EGGS-PTP framework exhibits a higher pruning overhead compared to baseline approaches, the cost remains acceptable within the broader context of LLMs deployment. Importantly, pruning is a one-time offline procedure, while the resulting sparsity enables substantial inference-time acceleration and memory savings.

Table 9: Pruning overhead (in seconds) for OPT-1.3b under different methods and sparsity patterns.

| Method | 2:4 | 4:8 |
|---|---|---|
| Magnitude | 5.95 | 2.96 |
| Wanda | 41.82 | 39.12 |
| RIA | 60.18 | 48.27 |
| EGGS-PTP | 85.11 | 60.63 |

## H SENSITIVITY ANALYSIS ON THE HYPERPARAMETER $B$

To showcase how $B$ affects model performance, we conduct a sensitivity test on LLaMA3.2 1B under the 2:4 sparsity pattern. We vary $B$ in the range $[2, 20]$ and report the WikiText2 perplexity of the resulting pruned models. The results are shown in Figure 5. Overall, the performance varies only mildly across different choices of $B$. Importantly, for all tested values, EGGS-PTP maintains a significant advantage over baseline methods (e.g., Magnitude, Wanda, and RIA).

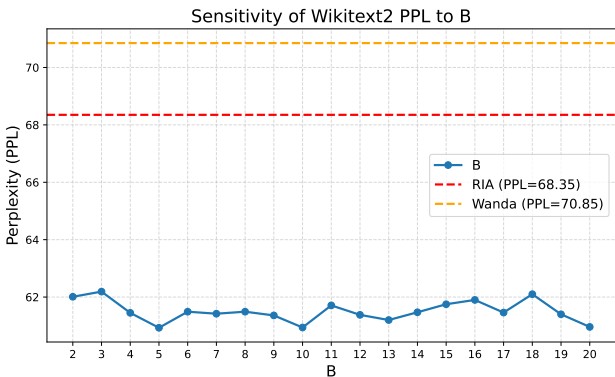

Figure 5: Sensitivity test of $B$ on LLaMA3.2-1b under $2 : 4$ sparsity pattern.

