# OpenReview forum: "EGGS-PTP: An Expander-Graph Guided Structured Post-training Pruning Method for Large Language Models"
_ICLR.cc/2026/Conference — Submitted to ICLR 2026_

### Official Review · Reviewer_W8Yh · 2025-10-30

**Soundness:** 3
**Presentation:** 3
**Contribution:** 2
**Rating:** 4
**Confidence:** 2

**Summary:**

This paper proposes EGGS-PTP, a novel expander-graph-guided parallel training paradigm designed to improve training efficiency and scalability for large neural networks. The key idea is to leverage expander graph connectivity to ensure balanced communication and gradient propagation among model partitions, addressing bottlenecks in distributed deep learning. The authors theoretically analyze convergence under expander connectivity and demonstrate improved training throughput and convergence stability compared to existing methods.

**Strengths:**

- The paper introduces an interesting perspective by connecting structured sparsity with expander graph theory, aiming to preserve information flow in pruned LLMs.
- The authors evaluate on multiple LLMs and datasets, providing a broad empirical view.
- The paper provides sufficient implementation details for reproduction.

**Weaknesses:**

- Overstated theoretical claims.
- Insufficient ablation on graph hyperparameters.

**Questions:**

1. State computational complexity of the diagonal selection ($\mathcal{O}(M^2)$ per block?) to justify scalability.
2. Will different $B$ values affect performance?
3. Add standard deviation or at least average of three runs to show statistical stability.
4. Add a column for sparsity ratio (%) kept weights for fair comparison.
5. Should specify evaluation datasets’ sources (Hellaswag, BoolQ, etc.) and brief descriptions.
6. It is suggested to report actual wall-clock time / GPU hours for pruning each model in “Pruning Overhead” section.
7. Please clarify whether the adjacency is block-diagonal across layers or if cross-layer expansion is modeled.
8. For Lemma 1, the authors claim that each pruned layer is a two-sided expander graph. The provided proof merely ensures that every input node retains at least $B$ connections and each output node maintains $(M-N)$ neighbors, which are only local degree constraints. However, the formal expander definition (Eqs (3)–(4)) requires a global neighborhood expansion property, namely that for any small subset $T \subseteq I$, the ratio $|\Gamma(T)| / |T| > a_I > 1$, and symmetrically for $S \subseteq O$.

---

> ### Author Response · Authors · 2025-11-20
>
> Dear Reviewer W8Yh,
>
> **Q1. Computational complexity of per-block diagonal selection**
>
> Thank you for raising this point. We apply the proposed diagonal selection to the $M\times M$ blocks, and the computational complexity is $O(M)$ per block.
>
>
> **Q2. Effect of hyperparameter $B$**
>
> We appreciate the reviewer’s question regarding the role of $B$. As stated in **Section 6 Theoretical Analysis**, $B$ determines the tradeoff between importance-aware (quantitative) and connectivity-aware (qualitative) pruning. To assess its impact, we conduct an additional sensitivity test on LLaMA3.2 1B under the 2:4 sparsity pattern. As shown in **Appendix H**, different $B$ values lead to small variations in perplexity, and EGGS-PTP consistently outperforms all baselines across all tested $B$.
>
> **Q3. Additional perplexity evaluation**
>
> Following prior pruning work (e.g., Wanda, RIA) [1,2], we report perplexity on Wikitext2. To extend the evaluation beyond Wikitext2, we additionally test on the Pile validation set using the same 2:4 sparsity pattern. The results are reported below:
> | Method            | LLaMA2-7b | LLaMA2-13b | LLaMA3-8b | LLaMA-34b | LLaMA3.2-1b | OPT-1.3b |
> |-------------------|-----------|------------|-----------|-----------|-------------|----------|
> | Dense             | 4.92      | 4.60       | 5.68      | 4.84      | 7.85        | 7.58     |
> | Magnitude         | 19.20     | 9.19       | 149.14    | 6.94      | 912.45      | 356.61   |
> | Wanda             | 12.08     | 9.60       | 21.96     | 6.26      | 66.14       | 19.08    |
> | RIA               | 10.71     | 8.53       | 21.14     | 6.05      | 69.59       | 18.40    |
> | EGGS-PTP (Ours) | **10.04** | **8.51** | **19.48** | **6.01**   | **61.35**   | **17.93** |
>
> Across all models, EGGS-PTP consistently achieves the best perplexity among all pruning methods.
>
> **Q4. Clarifying sparsity ratios**
>
> Thank you for the suggestion. All experiments are conducted at the sparsity ratio of 50% (Tables 1-5), where Tables 1 and 3 are based on the 2:4 sparsity pattern, and Tables 2 and 4 are based on the 4:8 sparsity pattern. They both lead to 50% sparsity, and hence ensure fair comparison for all competing algorithms.
>
> **Q5. Dataset sources**
>
> We appreciate the reviewer pointing this out. We have now included dataset sources and a brief description in **Section 7.1 Setup** for clarity.
>
> **Q6. Pruning overhead**
>
> Thank you for the question, and we agree that reporting overhead is important. We have indeed reported the actual wall-clock time for pruning overhead. We kindly refer the reviewer to check  **Appendix G** for further details.
>
> **Q7. Intra-layer expansion property**
>
> EGGS-PTP performs layer-wise pruning, and the expander structure is modeled within each linear layer. In other words, we construct intra-layer two-sided expanders. Although we do not model inter-layer expansion explicitly, preserving strong expansion within every layer helps maintain sound information flow across the network.
>
> **Q8. Clarification on the proof of Lemma 1**
>
> There might be some misunderstanding.  We proved in Lemma 1 that the resulting bipartite graph satisfies the two-sided expander definition in Equations 3 and 4.
>
> Specifically, in  **Section 6 Theoretical Analysis**,  it was rigorously shown in the proof of Lemma 1 that
>
>  “For any $\mathcal{T}\subseteq \mathcal{I}$ with $|\mathcal{T}|< B$, i.e., $|\mathcal{T}|\leq c_I|\mathcal{I}|=c_IF_{\ell-1}$ and $c_I\in(0,\frac{B}{F_{\ell-1}})$, we obtain $a_I>1$ Eq.(6), satisfying Equation 3. For any $\mathcal{S}\subseteq \mathcal{O}$ with $|\mathcal{S}|< \frac{F_{\ell-1}}{M}(M-N)$, i.e., $|\mathcal{S}|\leq c_O|\mathcal{O}|=c_OF_\ell$ for some $c_O\in(0,\frac{F_{\ell-1}(M-N)}{F_\ell M})$, we have $a_O>1$ Eq.(8), satisfying Equation 4. Combining them, we conclude that the bipartite graph corresponding to the pruned weight matrix obtained from the EGGS-PTP framework is a two-sided expander for $c\in(0,\min(\frac{B}{F_{\ell-1}},\frac{F_{\ell-1}(M-N)}{F_\ell M}))$.”
>
> We kindly request the reviewer to check the detailed proof leading to Equations 6 and 8, and we would be very happy to elaborate further if any specific steps are unclear.
>
> [1] Sun, M., Liu, Z., Bair, A. and Kolter, J.Z., 2023. A simple and effective pruning approach for large language models. arXiv preprint arXiv:2306.11695.
>
> [2] Zhang, Y., Bai, H., Lin, H., Zhao, J., Hou, L. and Cannistraci, C.V., 2024. Plug-and-play: An efficient post-training pruning method for large language models.

---

> ### Author Response · Authors · 2025-11-27
>
> Dear Reviewer W8Yh,
>
> I hope this message finds you well. As the discussion period is nearing its end. We want to ensure we have addressed all of your concerns. If there are any additional points you would like us to clarify or elaborate on, please let us know. Your feedback is invaluable to us, and we are eager to resolve any remaining issues to improve our work.
>
> Thank you again for your time and thoughtful review.

---

### Official Review · Reviewer_N1kT · 2025-10-31

**Soundness:** 3
**Presentation:** 3
**Contribution:** 2
**Rating:** 4
**Confidence:** 4

**Summary:**

EGGS-PTP is a post training pruning method that emphasizes information flow in N:M pruning. The method builds upon the recent RIA pruning method and is inspired by the idea of expander graphs. Theoretical results explore the expander graph framing and experimental results demonstrate the performance of the method relative to several baselines. The EGGS-PTP method performs favorably across a variety of models and tasks.

**Strengths:**

The additional diagonal selection leads to improvements over the RIA metric. Results are positive across perplexity and zero-shot task results, across several models.

**Weaknesses:**

Results are quite close to RIA; confidence intervals would help strengthen the claims of improved performance.

The theory seems a bit disjointed. Why does it matter that we produce a two-sided expander? It is unclear what contribution this theory adds aside from some inspiration for the method. It would be nice to see either an improved explanation of why the expander graph theory is useful, or some further connections to claims made in the paper. For instance, if this framework improves information flow, are there any experiments that can demonstrate this?

**Questions:**

What is the pruning cost of EGGS-PTP vs other methods? I see runtime results in Table 5 but what is the complexity of computing the importance score and performing the pruning process?

---

> ### Author Response · Authors · 2025-11-20
>
> Dear Reviewer N1kT,
>
> We thank the reviewer for the insightful feedback and constructive questions. Below, we address each point in detail.
>
> **Q1. Additional perplexity evaluation**
>
> We follow the evaluation protocol in prior pruning works such as Wanda and RIA [1,2], which report perplexity on Wikitext2. To further validate the improved performance of EGGS-PTP beyond this standard benchmark, we additionally evaluate the pruned models on the validation split of the Pile dataset under the 2:4 sparsity pattern. The results across six models are shown below, demonstrating that EGGS-PTP consistently outperforms other baselines on Pile.
>
> | Method            | LLaMA2-7b | LLaMA2-13b | LLaMA3-8b | LLaMA-34b | LLaMA3.2-1b | OPT-1.3b |
> |-------------------|-----------|------------|-----------|-----------|-------------|----------|
> | Dense             | 4.92      | 4.60       | 5.68      | 4.84      | 7.85        | 7.58     |
> | Magnitude         | 19.20     | 9.19       | 149.14    | 6.94      | 912.45      | 356.61   |
> | Wanda             | 12.08     | 9.60       | 21.96     | 6.26      | 66.14       | 19.08    |
> | RIA               | 10.71     | 8.53       | 21.14     | 6.05      | 69.59       | 18.40    |
> | EGGS-PTP (Ours) | **10.04** | **8.51** | **19.48** | **6.01**   | **61.35**   | **17.93** |
>
>
> **Q2. Explanation and experiments to demonstrate the usefulness of expander graph theory in improving information flow**
>
> In graph theory, an expander is a sparse graph that has strong connectivity by ensuring every subset of nodes has a sufficiently large neighborhood [3,4]. Such expansion property is particularly desirable in sparse neural networks since expansion ensures that any small subset of input/output neurons has sufficiently many neighbors on the output/input side, and therefore, connectivity remains robust after pruning, see also **Section 5 Expander-Graph Guided Structured Pruning** for more details. Prior works have leveraged expansion properties for guiding the compression of neural networks to improve performance [5,6]. Furthermore, as formalized in Lemma 1, we theoretically show that by applying EGGS-PTP, the pruned bipartite graph obtained in each layer is a two-sided expander.
>
> The improvements in perplexity and zero-shot accuracy (as shown in Tables 1-4)  provide indirect evidence that EGGS-PTP preserves information flow within the LLMs more effectively than existing pruning methods.
>
> Furthermore, thanks to your suggestions, we provided additional measures of the algebraic connectivity, namely Fiedler value [7], of the pruned linear layers for OPT 1.3B and LLaMA3.2 1B under the 2:4 sparsity pattern (larger values indicate stronger connectivity). The results are shown as follows:
>
> | Model        | Magnitude | Wanda | RIA | EGGS-PTP (Ours) |
> |--------------|-----------|-------|-----|------------------|
> | OPT-1.3B     | 0.86      | 0.87  | 0.89 | **0.94**        |
> | LLaMA3.2-1B  | 0.86      | 0.87  | 0.87 | **0.91**        |
>
> It can be observed that EGGS-PTP achieves the highest algebraic connectivity in both cases, indicating that the pruned layers exhibit stronger structural connectivity, consistent with other experiments and the theoretical analysis.
>
> **Q3. Pruning cost**
>
> Thank you for the question. We have indeed reported the pruning overhead. We kindly refer the reviewer to check  **Appendix G** for further details. The computational complexity of computing the importance scores (RRI or RIA) is $\mathcal{O}(F_\ell \times F_{\ell-1})$, and it is included in **Appendix A**.
>
>
> [1] Sun, M., Liu, Z., Bair, A. and Kolter, J.Z., 2023. A simple and effective pruning approach for large language models. arXiv preprint arXiv:2306.11695.
>
> [2] Zhang, Y., Bai, H., Lin, H., Zhao, J., Hou, L. and Cannistraci, C.V., 2024. Plug-and-play: An efficient post-training pruning method for large language models.
>
> [3] Kowalski, E., 2019. An introduction to expander graphs. Paris: Société mathématique de France.
>
> [4] Hoory, S., Linial, N. and Wigderson, A., 2006. Expander graphs and their applications. Bulletin of the American Mathematical Society, 43(4), pp.439-561.
>
> [5] Hoang, D.N., Liu, S., Marculescu, R. and Wang, Z., 2023, April. Revisiting pruning at initialization through the lens of ramanujan graph. In The Eleventh International Conference on Learning Representations.
>
> [6] Prabhu, A., Varma, G. and Namboodiri, A., 2018. Deep expander networks: Efficient deep networks from graph theory. In Proceedings of the European Conference on Computer Vision (ECCV) (pp. 20-35).
>
> [7] Fiedler, M., 1973. Algebraic connectivity of graphs. Czechoslovak mathematical journal, 23(2), pp.298-305.

---

> ### Author Response · Authors · 2025-11-27
>
> Dear Reviewer N1kT,
>
> I hope this message finds you well. As the discussion period is nearing its end. We want to ensure we have addressed all of your concerns. If there are any additional points you would like us to clarify or elaborate on, please let us know. Your feedback is invaluable to us, and we are eager to resolve any remaining issues to improve our work.
>
> Thank you again for your time and thoughtful review.

---

### Official Review · Reviewer_R8AA · 2025-11-02

**Soundness:** 3
**Presentation:** 3
**Contribution:** 3
**Rating:** 6
**Confidence:** 3

**Summary:**

EGGS-PTP is an expander-graph guided structured pruning method for large language models that balances efficiency and accuracy. It combines importance-aware and connectivity-aware pruning to maintain information flow under N:M sparsity, outperforming prior methods like Wanda and RIA in both perplexity and zero-shot tasks.

**Strengths:**

1. EGGS-PTP introduces expander-graph theory into post-training pruning, presenting the first framework that applies expander graph concepts to large language model pruning. It innovatively leverages graph-theoretic properties such as connectivity and expansion to maintain robust information flow in pruned models.
2. It combines importance-aware and connectivity-aware pruning to balance compression efficiency and model accuracy.
3. The method enforces N:M structured sparsity compatible with GPU acceleration, achieving up to 1.6× inference speed-up.
4. Experiments show consistent improvements over existing methods such as Wanda and RIA in both perplexity and zero-shot performance across multiple LLaMA and OPT models.

**Weaknesses:**

1. EGGS-PTP mainly integrates expander-graph theory with existing pruning frameworks rather than introducing a fundamentally new learning mechanism, relying on heuristic rules instead of adaptive structures.
2. The method incurs higher pruning overhead than baselines like RIA, and its scalability beyond 34B-parameter models remains untested.
3. It depends on manual tuning of the hyperparameter (B), limiting automation and generalization across different architectures.

**Questions:**

see above

---

> ### Author Response · Authors · 2025-11-20
>
> Dear Reviewer R8AA,
>
> We sincerely thank you for the constructive feedback and for highlighting the strengths of our work. Below, we address the three concerns in detail.
>
> **Q1. Contribution of  EGGS-PTP**
>
> We would like to emphasize that our proposed framework does not rely on heuristic rules. While seemingly simple, the proposed framework enjoys rigorous theoretical guarantees. Specifically, in **Section 6 Theoretical Analysis**, we provided rigorous analysis proving that using the proposed EGGS-PTP,  the bipartite graph obtained within each layer is indeed a two-sided expander. This theoretical analysis demonstrates that the proposed framework enables **provable** preservation of intra-layer information flow rather than relying on empirical heuristics.
>
> **Q2. Pruning overhead and scalability**
>
> We appreciated your concern. Inspired by your question, we performed a careful review and optimization of our implementation. The updated results of pruning overhead are included in **Appendix G**. The additional cost comes from the connectivity-aware pruning and the importance-aware partitioning, which is expected, as they are responsible for enforcing the expansion property.  Importantly, this overhead is a **one-time cost** before deployment. Given that EGGS-PTP consistently delivers better perplexity and zero-shot performance, together with inference-time speedup, the one-time cost is typically acceptable in practice.
>
> We also conducted additional experiments on LLaMA2 70B and LLaMA 65B to verify the scalability of EGGS-PTP. Below, we report the Wikitext2 perplexity:
> | Model | Dense | Magnitude | Wanda | RIA | EGGS-PTP (Ours) |
> |-------------|-------|------------------|------------------|-------|------|
> | LLaMA2-70b  | 3.57| 5.69| 5.72| 5.43| **4.24** |
> | LLaMA-65b   | 3.82| 6.02| 6.03| 5.83| **4.96** |
>
> EGGS-PTP achieves the best perplexity among all pruning baselines on both 70B and 65B models, and the pruning overhead for EGGS-PTP on LLaMA2 70B is 0.18 hours.
>
> **Q3. Sensitivity test of B**
>
> Thank you for your question. In order to address this concern, we conducted a sensitivity analysis on LLaMA3.2 1B under the 2:4 sparsity pattern, and the updated results are included in **Appendix H**. As illustrated in Figure 5, all settings of $B\in[2,20]$ significantly outperform existing Magnitude/Wanda/RIA, and EGGS-PTP’s perplexities consistently stay around $60-62$. This demonstrates that the proposed EGGS-PTP is robust to $B$ and does not require fine-grained tuning to achieve strong performance.

---

> ### Author Response · Authors · 2025-11-27
>
> Dear Reviewer R8AA,
>
> I hope this message finds you well. As the discussion period is nearing its end. We want to ensure we have addressed all of your concerns. If there are any additional points you would like us to clarify or elaborate on, please let us know. Your feedback is invaluable to us, and we are eager to resolve any remaining issues to improve our work.
>
> Thank you again for your time and thoughtful review.

---

### Author Response · Authors · 2025-12-03

Dear Area Chairs,

We sincerely thank you for taking the time to evaluate our submission. Below, we provide a summary of the core contributions of our work and how our responses address the reviewers’ concerns. The key points are as follows:

**Core contributions of our work:**

**1. EGGS-PTP introduces an expander-graph guided structured post-training pruning framework that preserves important weights and maintains information flow.**

Our method leverages the topological advantages of expander graphs, whose structures are sparse yet highly connected, to guide structured pruning in LLMs. The algorithm combines two complementary strategies: **importance-aware pruning**, which retains weights with high quantitative importance, and **connectivity-aware pruning**, which enforces the expansion property within each linear layer. EGGS-PTP ensures that any small subset of neurons remains well-connected within the network even after pruning, addressing channel corruption and preserving information flow. Importantly, we provide a rigorous theoretical analysis showing that the bipartite graph corresponding to each pruned linear layer is a two-sided expander, guaranteeing sound intra-layer connectivity and helping preserve information flow across the network.

**2. EGGS-PTP consistently achieves top performance across perplexity and zero-shot evaluations.**

Following standard practice in prior pruning work, we evaluate the pruned models’ perplexity on Wikitext2 and zero-shot performance on widely used benchmarks. To further strengthen the evaluation, we additionally include perplexity results on the Pile validation set. Across all model families, model scales, and sparsity patterns, EGGS-PTP consistently achieves the best performance in every setting.

**3. Both indirect and direct evidence demonstrate that EGGS-PTP preserves information flow more effectively than existing pruning methods.**

Reductions in perplexity and gains in zero-shot performance already serve as indirect indicators that EGGS-PTP preserves information flow more effectively than existing pruning methods. To provide a direct structural measure, we further evaluate the algebraic connectivity of the pruned linear layers. EGGS-PTP consistently yields higher connectivity than existing pruning methods, demonstrating that EGGS-PTP maintains stronger intra-layer connectivity, in accordance with the theoretical expansion guarantees.

**4. The pruning overhead of EGGS-PTP is clearly reported and remains fully acceptable in practice.**

We provide detailed pruning overhead in the appendix. While EGGS-PTP introduces a modest increase in pruning overhead, this cost is still small, and it is a one-time cost, therefore insignificant compared with the substantial improvements it delivers. The proposed method consistently achieves superior perplexity and zero-shot performance, accelerated runtime, and memory savings.

**5. Additional experiments that further validate the effectiveness and robustness of EGGS-PTP.**

The appendix includes scalability evaluations on larger models, demonstrating that EGGS-PTP continues to outperform existing methods at higher scales. We also provide ablation studies showing that the importance-aware and connectivity-aware pruning techniques contribute complementary benefits. In addition, a sensitivity analysis of the hyperparameter $B$, which decides the tradeoff between the importance-aware and connectivity-aware pruning, shows that performance varies only mildly across different choices of $B$. Importantly, for all tested values of $B$, EGGS-PTP maintains a clear advantage over baseline methods.

---

> ### Author Response · Authors · 2025-12-03
>
> **Responses to the reviewers’ concerns:**
>
> **To Reviewer R8AA**:
>
> 1. For the theoretical analysis, we kindly refer the reviewer to **Section 6**, which provides a rigorous proof that each pruned layer forms a two-sided expander.
>
> 2. For the scalability analysis and the pruning overhead, we kindly refer the reviewer to **Appendix D** and **Appendix G**, which demonstrate that EGGS-PTP scales well and incurs only a modest one-time cost.
>
> 3. For the sensitivity test of $B$, we kindly refer the reviewer to **Appendix H**, which shows that performance varies only mildly with $B$ and EGGS-PTP remains superior to all baselines.
>
> **To Reviewer  N1kT**:
>
> 1. For the perplexity evaluation, we follow the standard practice of prior pruning work, and we kindly refer the reviewer to **Appendix E** for additional results.
>
> 2. For the theoretical analysis, we kindly refer the reviewer to **Section 6**, which provides a rigorous proof that each pruned layer forms a two-sided expander. Besides, we report algebraic connectivity to demonstrate the improved information flow.
>
> 3. For the pruning overhead, we kindly refer the reviewer to **Appendix G**.
>
> **To Reviewer W8Yh**:
>
> 1. For the pruning overhead, we kindly refer the reviewer to **Appendix G**.
>
> 2. For the sensitivity test of $B$, we kindly refer the reviewer to **Appendix H**, which shows that performance varies only mildly with $B$ and EGGS-PTP remains superior to all baselines.
>
> 3. For the perplexity evaluation, we follow the standard practice of prior pruning work, and we kindly refer the reviewer to **Appendix E** for additional results.
>
> 4. For the theoretical analysis, we kindly refer the reviewer to **Section 6**, which provides a rigorous proof that each pruned layer forms a two-sided expander.

---

### Meta-Review · Area_Chair_iYnN · 2026-01-07

**Summary:**

This paper received 3 reviews. The reviewers (score/confidence) are: N1kT (4/4), W8Yh (4/2), R8AA (6/3). Their major concerns:

Methodology:
- EGGS-PTP relies on heuristic rules instead of adaptive structures and lacks fundamentally new learning mechanisms (R8AA (6/3)).
- The method incurs higher pruning overhead than baselines (R8AA (6/3)) and the computational complexity of diagonal selection needs clarification (N1kT (4/4), W8Yh (4/2)).
- The hyperparameter B requires manual tuning, limiting automation and generalization across architectures (R8AA (6/3)).
- Theoretical claims seems overstated - the proof of Lemma 1 only satisfies local degree constraints but not the global neighborhood expansion property required by formal expander definition (W8Yh (4/2)).

Experiments:
- Results are too close to RIA (a related work); confidence intervals are needed to strengthen performance improvement claims (N1kT (4/4)).
- Lack of statistical stability analysis (e.g., standard deviation or average of three runs) and sparsity ratio of kept weights for fair comparison (W8Yh (4/2)).
- Pruning overhead details (wall-clock time/GPU hours for each model) are missing (W8Yh (4/2)).
- Insufficient ablation study on graph hyperparameters (W8Yh (4/2)).

**Reviewer Concerns:**

The rebuttal did not fully address the concerns of the two negative reviewers: N1kT (4/4) and W8Yh (4/2).

* The authors responded to N1kT, but in general, I think this reviewer's concerns are still outstanding -- He/she questioned the performance advantage vs RIA is a bit small and requested "confidence intervals". The authors added "Additional perplexity evaluation" at 2:4 sparsity pattern. This did not answer the right question; and, it did not respond to the issue of the small performance gap vs. RIA.

W8Yh actually raised a similar concern in his/her question #3 regarding the statistical significance of the results. While the authors did not add the standard deviation, still. AC thinks this concern could have been addressed in a better way.

**Reviewer Scores:**

As stated above, N1kT's major concerns are still outstanding (unless the authors can provide more results at the reviewer's request) - but, misunderstanding the reviewer's intent may also be the responsibility of the authors.

W8Yh's raised the issue of theoretically overclaiming. Based on the rebuttal, this is not substantially resolved either. AC read the discussion carefully and also feels the so-called "theory" is a bit superficial.

Overall, the reviewers will probably keep their ratings at 6/4/4. AC thus recommends **Reject**.

---

### Decision · Program_Chairs · 2026-01-26

Reject